# Mind the Nuisance: Gaussian Process Classification using Privileged Noise

**Daniel Hernández-Lobato**
Universidad Autónoma de Madrid
Madrid, Spain
daniel.hernandez@uam.es

**Viktoriia Sharmanska**
IST Austria
Klosterneuburg, Austria
vsharman@ist.ac.at

**Kristian Kersting**
TU Dortmund
Dortmund, Germany
first.last@cs.tu-dortmund.de

**Christoph H. Lampert**
IST Austria
Klosterneuburg, Austria
chl@ist.ac.at

**Novi Quadrianto**
SMiLe CLiNiC, University of Sussex
Brighton, United Kingdom
n.quadrianto@sussex.ac.uk

## Abstract

The *learning with privileged information* setting has recently attracted a lot of attention within the machine learning community, as it allows the integration of additional knowledge into the training process of a classifier, even when this comes in the form of a data modality that is not available at test time. Here, we show that privileged information can naturally be treated as noise in the latent function of a Gaussian process classifier (GPC). That is, in contrast to the standard GPC setting, the latent function is not just a nuisance but a feature: it becomes a natural measure of confidence about the training data by modulating the slope of the GPC probit likelihood function. Extensive experiments on public datasets show that the proposed GPC method using privileged noise, called GPC+, improves over a standard GPC without privileged knowledge, and also over the current state-of-the-art SVM-based method, SVM+. Moreover, we show that advanced neural networks and deep learning methods can be compressed as privileged information.

## 1 Introduction

Prior knowledge is a crucial component of any learning system as without a form of prior knowledge learning is provably impossible [1]. Many forms of integrating prior knowledge into machine learning algorithms have been developed: as a preference of certain prediction functions over others, as a Bayesian prior over parameters, or as additional information about the samples in the training set used for learning a prediction function. In this work, we rely on the last of these setups, adopting Vapnik and Vashist's *learning using privileged information (LUPI)*, see e.g. [2, 3]: *we want to learn a prediction function, e.g. a classifier, and in addition to the main data modality that is to be used for prediction, the learning system has access to additional information about each training example.*

This scenario has recently attracted considerable interest within the machine learning community because it reflects well the increasingly relevant situation of *learning as a service*: an expert trains a machine learning system for a specific task on request from a customer. Clearly, in order to achieve the best result, the expert will use all the information available to him or her, not necessarily just the

information that the system itself will have access to during its operation after deployment. Typical scenarios for learning as a service include visual inspection tasks, in which a classifier makes real-time decisions based on the input from its sensor, but at training time, additional sensors could be made use of, and the processing time per training example plays less of a role. Similarly, a classifier built into a robot or mobile device operates under strong energy constraints, while at training time, energy is less of a problem, so additional data can be generated and made use of. A third scenario is when the additional data is *confidential*, as e.g. in health care applications. Specifically, a diagnosis system may be improved when more information is available at training time, e.g., specific blood tests, genetic sequences, or drug trials, for the subjects that form the training set. However, the same data may not be available at test time, as obtaining it could be impractical, unethical, or illegal.

We propose a novel method for using privileged information based on the framework of Gaussian process classifiers (GPCs). The privileged data enters the model in form of a latent variable, which modulates the noise term of the GPC. Because the noise is integrated out before obtaining the final model, the privileged information is only required at training time, not at prediction time. The most interesting aspect of the proposed model is that by this procedure, the influence of the privileged information becomes very interpretable: its role is to model the confidence that the GPC has about any training example, which can be directly read off from the slope of the probit likelihood. Instances that are *easy* to classify by means of their privileged data cause a faster increasing probit, which means the GP trusts the training example and tried to fit it well. Instances that are *hard* to classify result in a slowly increasing slope, so that the GPC considers them less reliable and does not put a lot of effort in fitting their label well. Our experiments on multiple datasets show that this procedure leads not just to more interpretable models, but also to better prediction accuracy.

**Related work:** The LUPI framework was originally proposed by Vapnik and Vashist [2], inspired by a thought-experiment: *when training a soft-margin SVM, what if an* oracle *would provide us with the optimal values of the slack variables?* As it turns out, this would actually provably reduce the amount of training data needed, and consequently, Vapnik and Vashist proposed the *SVM+* classifier that uses privileged data to predict values for the slack variables, which led to improved performance on several categorisation tasks and found applications, e.g., in finance [4]. This setup was subsequently improved, by a faster training algorithm [5], better theoretical characterisation [3], and it was generalised, e.g., to the *learning to rank* setting [6], *clustering* [7], *metric learning* [8] and *multi-class data classification* [9]. Recently, however, it was shown that the main effect of the SVM+ procedure is to assign a data-dependent weight to each training example in the SVM objective [10].

The proposed method, GPC+, constitutes the first Bayesian treatment of classification using privileged information. The resulting *privileged noise* approach is related to input-modulated noise commonly done in the regression task, where several Bayesian treatments of *heteroscedastic regression* using GPs have been proposed. Since the predictive density and marginal likelihood are no longer analytically tractable, most works deal with approximate inference, *i.e.*, techniques such as Markov Chain Monte Carlo [11], maximum a posteriori [12], and variational Bayes [13]. To our knowledge, however, there is no prior work on *heteroscedastic classification* using GPs — we will elaborate the reasons in Section 2.1 — and this work is the first to develop approximate inference based on expectation propagation for the heteroscedastic noise case in the context of classification.

## 2 GPC+: Gaussian process classification with privileged noise

For self-consistency we first review the GPC model [14] with an emphasis on the noise-corrupted latent Gaussian process view. Then, we show how to treat privileged information as heteroscedastic noise in this process. An elegant aspect of this view is how the privileged noise is able to distinguish between *easy* and *hard* samples and to re-calibrate the uncertainty on the class label of each instance.

### 2.1 Gaussian process classifier with noisy latent process

Consider a set of $N$ input-output data points or samples $\mathcal{D} = \{(\mathbf{x}_1, y_1), \ldots, (\mathbf{x}_N, y_N)\} \subset \mathbb{R}^d \times \{0, 1\}$. Assume that the class label $y_i$ of the sample $\mathbf{x}_i$ has been generated as $y_i = \mathbb{I}[\, \tilde{f}(\mathbf{x}_i) \geq 0 \,]$, where $\tilde{f}(\cdot)$ is a *noisy* latent function and $\mathbb{I}[\cdot]$ is the Iverson's bracket notation, *i.e.*, $\mathbb{I}[\, P \,] = 1$ when the condition $P$ is true, and 0 otherwise. Induced by the label generation process, we adopt the

following form of likelihood function for $\tilde{\mathbf{f}} = (\tilde{f}(\mathbf{x}_1), \ldots, \tilde{f}(\mathbf{x}_N))^\top$:

$$\Pr(\mathbf{y}|\tilde{\mathbf{f}}, X = (\mathbf{x}_1, \ldots, \mathbf{x}_N)^\top) = \prod_{n=1}^{N} \Pr(y_n = 1|\mathbf{x}_n, \tilde{f}) = \prod_{n=1}^{N} \mathbb{I}[\,\tilde{f}(\mathbf{x}_n) \geq 0\,], \qquad (1)$$

where $\tilde{f}(\mathbf{x}_n) = f(\mathbf{x}_n) + \epsilon_n$ with $f(\mathbf{x}_n)$ being the *noise-free* latent function. The noise term $\epsilon_n$ is assumed to be independent and normally distributed with zero mean and variance $\sigma^2$, that is $\epsilon_n \sim \mathcal{N}(\epsilon_n|0, \sigma^2)$. To make inference about $\tilde{f}(\mathbf{x}_n)$, we need to specify a prior over this function. We proceed by imposing a zero mean Gaussian process prior [14] on the noise-free latent function, that is $f(\mathbf{x}_n) \sim \mathcal{GP}(0, k(\mathbf{x}_n, \cdot))$ where $k(\cdot, \cdot)$ is a positive-definite kernel function [15] that specifies prior properties of $f(\cdot)$. A typical kernel function that allows for non-linear smooth functions is the squared exponential kernel $k_f(\mathbf{x}_n, \mathbf{x}_m) = \theta \exp(-\frac{1}{2l} \|\mathbf{x}_n - \mathbf{x}_m\|_{\ell_2}^2)$, where $\theta$ controls the prior amplitude of $f(\cdot)$ and $l$ controls its prior smoothness. The prior and the likelihood are combined using Bayes' rule to get the posterior of $\tilde{f}(\cdot)$. Namely, $\Pr(\tilde{\mathbf{f}}|X, \mathbf{y}) = \Pr(\mathbf{y}|\tilde{\mathbf{f}}, X)\Pr(\tilde{\mathbf{f}})/\Pr(\mathbf{y}|X)$.

We can simplify the above noisy latent process view by integrating out the noise term $\epsilon_n$ and writing down the individual likelihood at sample $\mathbf{x}_n$ in terms of the noise-free latent function $f(\cdot)$. Namely,

$$\Pr(y_n = 1|\mathbf{x}_n, f) = \int \mathbb{I}[\tilde{f}(\mathbf{x}_n) \geq 0]\mathcal{N}(\epsilon_n|0, \sigma^2)d\epsilon_n = \Phi_{(0,\sigma^2)}(f(\mathbf{x}_n)), \qquad (2)$$

where we have used that $\tilde{f}(\mathbf{x}_n) = f(\mathbf{x}_n) + \epsilon_n$ and $\Phi_{(\mu,\sigma^2)}(\cdot)$ is a Gaussian cumulative distribution function (CDF) with mean $\mu$ and variance $\sigma^2$. Typically the standard Gaussian CDF is used, that is $\Phi_{(0,1)}(\cdot)$, in the likelihood of (2). Coupled with a Gaussian process prior on the latent function $f(\cdot)$, this results in the widely adopted noise-free latent Gaussian process view with probit likelihood. The equivalence between a noise-free latent process with probit likelihood and a noisy latent process with step-function likelihood is widely known [14]. It is also widely accepted that the function $\tilde{f}(\cdot)$ (or the function $f(\cdot)$) is a *nuisance* function as we do not observe its value and its sole purpose is for a convenient formulation of the model [14]. However, in this paper, we show that by using privileged information as the noise term, the latent function $\tilde{f}$ now plays a crucial role. The latent function with privileged noise adjusts the slope transition in the Gaussian CDF to be *faster* or *slower* corresponding to more *certainty* or more *uncertainty* about the samples in the original input space.

## 2.2 Introducing privileged information into the nuisance function

In the learning under privileged information (LUPI) paradigm [2], besides input data points $\{\mathbf{x}_1, \ldots, \mathbf{x}_N\}$ and associated labels $\{y_1, \ldots, y_N\}$, we are given additional information $\mathbf{x}_n^* \in \mathbb{R}^{d^*}$ about each training instance $\mathbf{x}_n$. However, this privileged information will not be available for unseen test instances. Our goal is to exploit the additional data $\mathbf{x}^*$ to influence our choice of the latent function $\tilde{f}(\cdot)$. This needs to be done while making sure that the function does not directly use the privileged data as input, as it is simply not available at test time. We achieve this naturally by treating the privileged information as a heteroscedastic (input-dependent) noise in the latent process.

Our classification model with privileged noise is then as follows:

$$\text{Likelihood model}: \Pr(y_n = 1|\mathbf{x}_n, \tilde{f}) = \mathbb{I}[\,\tilde{f}(\mathbf{x}_n) \geq 0\,], \quad \text{where} \quad \mathbf{x}_n \in \mathbb{R}^d \qquad (3)$$

$$\text{Assume}: \tilde{f}(\mathbf{x}_n) = f(\mathbf{x}_n) + \epsilon_n \qquad (4)$$

$$\text{Privileged noise model}: \epsilon_n \overset{i.i.d.}{\sim} \mathcal{N}(\epsilon_n|0, z(\mathbf{x}_n^*) = \exp(g(\mathbf{x}_n^*))), \quad \text{where} \quad \mathbf{x}_n^* \in \mathbb{R}^{d^*} \qquad (5)$$

$$\text{GP prior model}: f(\mathbf{x}_n) \sim \mathcal{GP}(0, k_f(\mathbf{x}_n, \cdot)) \quad \text{and} \quad g(\mathbf{x}_n^*) \sim \mathcal{GP}(0, k_g(\mathbf{x}_n^*, \cdot)). \qquad (6)$$

In the above, the function $\exp(\cdot)$ is needed to ensure positivity of the noise variance. The term $k_g(\cdot, \cdot)$ is a positive-definite kernel function that specifies the prior properties of another latent function $g(\cdot)$, which is evaluated in the privileged space $\mathbf{x}^*$. Crucially, the noise term $\epsilon_n$ is now *heteroscedastic*, that is, it has a different variance $z(\mathbf{x}_n^*)$ at each input point $\mathbf{x}_n$. This is in contrast to the standard GPC approach discussed in Section 2.1 where the noise term is homoscedastic, $\epsilon_n \sim \mathcal{N}(\epsilon_n|0, z(\mathbf{x}_n^*) = \sigma^2)$. An input-dependent noise term is very common in regression tasks with continuous output values $y_n \in \mathbb{R}$, resulting in heteroscedastic regression models, which have been proven more flexible in numerous applications as already touched upon in the section on related work. However, to our knowledge, there is no prior work on *heteroscedastic classification* models. This is not surprising as the nuisance view of the latent function renders a flexible input-dependent noise point-less.

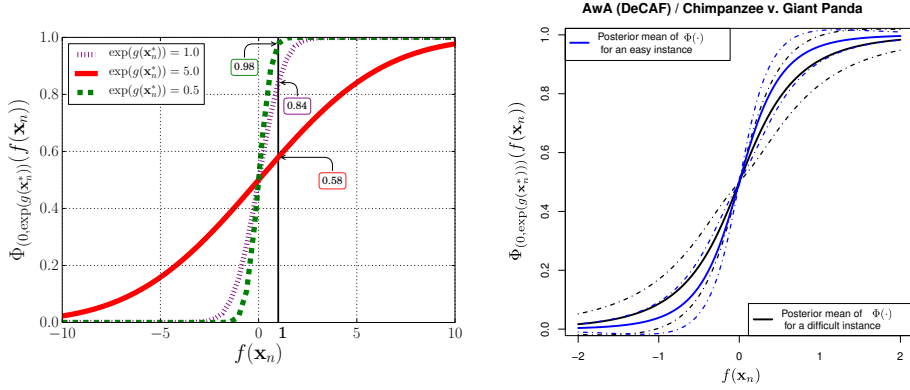

Figure 1: Effects of privileged noise on the nuisance function. **(Left)** On synthetic data. Suppose for an input $\mathbf{x}_n$, the latent function value is $f(\mathbf{x}_n) = 1$. Now also assume that the associated privileged information $\mathbf{x}_n^*$ for the $n$-th data point deems the sample as *difficult*, say $\exp(g(\mathbf{x}_n^*)) = 5.0$. Then the likelihood will reflect this uncertainty $\Pr(y_n = 1|f, g, \mathbf{x}_n, \mathbf{x}_n^*) = 0.58$. In contrast, if the associated privileged information considers the sample as *easy*, say e.g. $\exp(g(\mathbf{x}_n^*)) = 0.5$, the likelihood is very certain $\Pr(y_n = 1|f, g, \mathbf{x}_n, \mathbf{x}_n^*) = 0.98$. **(Right)** On real data taken from our experiments in Sec. 4. The posterior means of the $\Phi(\cdot)$ function (solid) and its 1-standard deviation confidence interval (dash-dot) for easy (blue) and difficult (black) instances of the Chimpanzee v. Giant Panda binary task on the Animals with Attributes (AwA) dataset. (Best viewed in color).

In the context of privileged information heteroscedastic classification is a very sensible idea, which is best illustrated when investigating the effect of privileged information in the equivalent formulation of a noise free latent process, *i.e.*, when one integrates out the privileged input-dependent noise term:

$$\Pr(y_n = 1|\mathbf{x}_n, \mathbf{x}_n^*, f, g) = \int \mathbb{I}[\ \tilde{f}(\mathbf{x}_n) \geq 0\ ]\mathcal{N}(\epsilon_n|0, \exp(g(\mathbf{x}_n^*))d\epsilon_n$$

$$= \Phi_{(0,\exp(g(\mathbf{x}_n^*)))}(f(\mathbf{x}_n)) = \Phi_{(0,1)}(f(\mathbf{x}_n)/\sqrt{\exp(g(\mathbf{x}_n^*))}). \qquad (7)$$

This equation shows that the privileged information adjusts the slope transition of the Gaussian CDF through the latent function $g(\cdot)$. For difficult samples the latent function $g(\cdot)$ will be high, the slope transition will be slower, and thus more uncertainty will be in the likelihood $\Pr(y_n = 1|\mathbf{x}_n, \mathbf{x}_n^*, f, g)$. For easy samples, however, $g(\cdot)$ will be low, the slope transition will be faster, and thus less uncertainty will be in the likelihood term. This behaviour is illustrated in Figure 1. For non-informative samples in the privileged space, the value of $g$ for those samples should be equal to a global noise value, as in a standard GPC. Thus, privileged information should in principle never hurt. Proving this theoretically is, however, an interesting and challenging research direction. Experimentally, however, we observe in the section on experiments the scenario described.

### 2.3 Posterior and prediction on test data

Define $\mathbf{g} = (g(\mathbf{x}_1^*), \ldots, g(\mathbf{x}_n^*))^{\mathrm{T}}$ and $\mathbf{X}^* = (\mathbf{x}_1^*, \ldots, \mathbf{x}_n^*)^{\mathrm{T}}$. Given the likelihood $\Pr(\mathbf{y}|\mathbf{X}, \mathbf{X}^\star, \mathbf{f}, \mathbf{g}) = \prod_{n=1}^{N} \Pr(y_n = 1|f, g, \mathbf{x}_n, \mathbf{x}_n^*)$ with the individual term $\Pr(y_n|f, g, \mathbf{x}_n, \mathbf{x}_n^*)$ given in (7) and the Gaussian process priors on functions, the posterior for $\mathbf{f}$ and $\mathbf{g}$ is:

$$\Pr(\mathbf{f}, \mathbf{g}|\mathbf{y}, \mathbf{X}, \mathbf{X}^\star) = \frac{\Pr(\mathbf{y}|\mathbf{X}, \mathbf{X}^\star, \mathbf{f}, \mathbf{g})\Pr(\mathbf{f})\Pr(\mathbf{g})}{\Pr(\mathbf{y}|\mathbf{X}, \mathbf{X}^\star)}, \qquad (8)$$

where $\Pr(\mathbf{y}|\mathbf{X}, \mathbf{X}^\star)$ can be maximised with respect to a set of hyper-parameter values such as the amplitude $\theta$ and the smoothness $l$ of the kernel functions [14]. For a previously unseen test point $\mathbf{x}_{\mathrm{new}} \in \mathbb{R}^d$, the predictive distribution for its label $y_{\mathrm{new}}$ is given as:

$$\Pr(y_{\mathrm{new}} = 1|\mathbf{y}, \mathbf{X}, \mathbf{X}^\star) = \int \mathbb{I}[\ \tilde{f}(\mathbf{x}_{\mathrm{new}}) \geq 0\ ]\Pr(f_{\mathrm{new}}|\mathbf{f})\Pr(\mathbf{f}, \mathbf{g}|\mathbf{y}, \mathbf{X}, \mathbf{X}^\star)d\mathbf{f}d\mathbf{g}df_{\mathrm{new}}, \qquad (9)$$

where $\Pr(f_{\mathrm{new}}|\mathbf{f})$ is a Gaussian conditional distribution. We note that in (9) we do not consider the privileged information $\mathbf{x}_{\mathrm{new}}^\star$ associated to $\mathbf{x}_{\mathrm{new}}$. The interpretation is that we consider homoscedastic

noise at test time. This is a reasonable approach as there is no additional information for increasing or decreasing our confidence in the newly observed data $\mathbf{x}_{\text{new}}$. Finally, we predict the label for a test point via Bayesian decision theory: the label being predicted is the one with the largest probability.

## 3 Expectation propagation with numerical quadrature

Unfortunately, as for most interesting Bayesian models, inference in the GPC+ model is very challenging. Already in the homoscedastic case, the predictive density and marginal likelihood are not tractable. Here, we therefore adapt Minka's expectation propagation (EP) [16] with numerical quadrature for approximate inference. Our choice is supported on the fact that EP is the preferred method for approximate inference in GPCs, in terms of accuracy and computational cost [17, 18].

Consider the joint distribution of $\mathbf{f}$, $\mathbf{g}$ and $\mathbf{y}$, $\Pr(\mathbf{y}|\mathbf{X}, \mathbf{X}^*, \mathbf{f}, \mathbf{g})\Pr(\mathbf{f})\Pr(\mathbf{g})$, where $\Pr(\mathbf{f})$ and $\Pr(\mathbf{g})$ are Gaussian process priors and the likelihood $\Pr(\mathbf{y}|\mathbf{X}, \mathbf{X}^*, \mathbf{f}, \mathbf{g})$ equals $\prod_{n=1}^{N} \Pr(y_n|\mathbf{x}_n, \mathbf{x}_n^*, f, g)$, with $\Pr(y_n|\mathbf{x}_n, \mathbf{x}_n^*, f, g)$ given by (7). EP approximates each non-normal factor in this distribution by an un-normalised bi-variate normal distribution of $f$ and $g$ (we assume independence between $f$ and $g$). The only non-normal factors are those of the likelihood, which are approximated as:

$$\Pr(y_n|\mathbf{x}_n, \mathbf{x}_n^*, f, g) \approx \overline{\gamma}_n(f, g) = \overline{z}_n \mathcal{N}(f(\mathbf{x}_n)|\overline{m}_f, \overline{v}_f)\mathcal{N}(g(\mathbf{x}_n^*)|\overline{m}_g, \overline{v}_g), \qquad (10)$$

where the parameters with the super-script $^-$ are to be found by EP. The posterior approximation $\mathcal{Q}$ computed by EP results from normalising with respect to $\mathbf{f}$ and $\mathbf{g}$ the EP approximate joint. That is, $\mathcal{Q}$ is obtained by replacing each likelihood factor by the corresponding approximate factor $\overline{\gamma}_n$:

$$\Pr(\mathbf{f}, \mathbf{g}|\mathbf{X}, \mathbf{X}^*, \mathbf{y}) \approx \mathcal{Q}(\mathbf{f}, \mathbf{g}) := Z^{-1}[\prod_{n=1}^{N} \overline{\gamma}(f, g)]\Pr(\mathbf{f})\Pr(\mathbf{g}), \qquad (11)$$

where $Z$ is a normalisation constant that approximates the model evidence, $\Pr(\mathbf{y}|\mathbf{X}, \mathbf{X}^*)$. The normal distribution belongs to the exponential family of probability distributions and is closed under the product and division. It is hence possible to show that $\mathcal{Q}$ is the product of two multi-variate normals [19]. The first normal approximates the posterior for $\mathbf{f}$ and the second the posterior for $\mathbf{g}$.

EP tries to fix the parameters of $\overline{\gamma}_n$ so that it is similar to the exact factor $\Pr(y_n|\mathbf{x}_n, \mathbf{x}_n^*, f, g)$ in regions of high posterior probability [16]. For this, EP iteratively updates each $\overline{\gamma}_n$ until convergence to minimise $\text{KL}\left(\Pr(y_n|\mathbf{x}_n, \mathbf{x}_n^*, f, g)\mathcal{Q}^{\text{old}}/Z_n||\mathcal{Q}\right)$, where $\mathcal{Q}^{\text{old}}$ is a normal distribution proportional to $\left[\prod_{n'\neq n} \overline{\gamma}_{n'}\right]\Pr(\mathbf{f})\Pr(\mathbf{g})$ with all variables different from $f(\mathbf{x}_n)$ and $g(\mathbf{x}_n^*)$ marginalised out, $Z_n$ is simply a normalisation constant and $\text{KL}(\cdot||\cdot)$ denotes the Kullback-Leibler divergence between probability distributions. Assume $\mathcal{Q}^{\text{new}}$ is the distribution minimising the previous divergence. Then, $\overline{\gamma}_n \propto \mathcal{Q}^{\text{new}}/\mathcal{Q}^{\text{old}}$ and the parameter $\overline{z}_n$ of $\overline{\gamma}_n$ is fixed to guarantee that $\overline{\gamma}_n$ integrates the same as the exact factor with respect to $\mathcal{Q}^{\text{old}}$. The minimisation of the KL divergence involves matching expected sufficient statistics (mean and variance) between $\Pr(y_n|\mathbf{x}_n, \mathbf{x}_n^*, f, g)\mathcal{Q}^{\text{old}}/Z_n$ and $\mathcal{Q}^{\text{new}}$. These expectations can be obtained from the derivatives of $\log Z_n$ with respect to the (natural) parameters of $\mathcal{Q}^{\text{old}}$ [19]. Unfortunately, the computation of $\log Z_n$ in closed form is intractable. We show here that it can be approximated by a *one dimensional quadrature*. Denote by $m_f$, $v_f$, $m_g$ and $v_g$ the means and variances of $\mathcal{Q}^{\text{old}}$ for $f(\mathbf{x}_n)$ and $g(\mathbf{x}_n^*)$, respectively. Then,

$$Z_n = \int \Phi_{(0,1)}\left(y_n m_f/\sqrt{v_f + \exp(g(\mathbf{x}_n^*))}\right) \mathcal{N}(g(\mathbf{x}_n^*)|m_g, v_g)dg(\mathbf{x}_n^*). \qquad (12)$$

Thus, EP only requires five quadratures to update each $\overline{\gamma}_n$. One to compute $\log Z_n$ and four extras to compute its derivatives with respect to $m_f$, $v_f$, $m_g$ and $v_g$. After convergence, $\mathcal{Q}$ can be used to approximate predictive distributions and the normalisation constant $Z$ can be maximised to find good values for the model's hyper-parameters. In particular, it is possible to compute the gradient of $Z$ with respect to the parameters of the Gaussian process priors for $\mathbf{f}$ and $\mathbf{g}$ [19]. An R language implementation of GPC+ using EP for approximate inference is found in the supplementary material.

## 4 Experiments

We investigate the performance of GPC+. To this aim we considered three types of binary classification tasks corresponding to different privileged information using two real-world datasets: *Attribute Discovery* and *Animals with Attributes*. We detail these experiments in turn in the following sections.

**Methods:** We compared our proposed `GPC+` method with the well-established LUPI method based on SVM, `SVM+` [5]. As a reference, we also fit standard GP and SVM classifiers when learning on the original space $\mathbb{R}^d$ (`GPC` and `SVM` baselines). For *all four* methods, we used a squared exponential kernel with amplitude parameter $\theta$ and smoothness parameter $l$. For simplicity, we set $\theta = 1.0$ in all cases. There are two hyper-parameters in GPC (smoothness parameter $l$ and noise variance $\sigma^2$) and also two in GPC+ (smoothness parameters $l$ of kernel $k_f(\cdot, \cdot)$ and of kernel $k_g(\cdot, \cdot)$). In GPC and GPC+, we used type II-maximum likelihood for finding all hyper-parameters. SVM has two knobs, *i.e.*, smoothness and regularisation, and SVM+ has four knobs, two smoothness and two regularisation parameters. In SVM we used a grid search guided by cross-validation to set all hyper-parameters. However, this procedure was too expensive for finding the best parameters in SVM+. Thus, we used the performance on a separate validation set to guide the search. This means that we give a competitive advantage to SVM+ over the other methods, which do not use the validation set.

**Evaluation metric:** To evaluate the performance of each method we used the classification error measured on an independent test set. We performed 100 repeats of all the experiments to get the better statistics of the performance and we report the mean and the standard deviation of the error.

### 4.1 Attribute discovery dataset

The data set was collected from a website that aggregates product data from a variety of e-commerce sources and includes both images and associated textual descriptions [20]. The images and texts are grouped into 4 broad shopping categories: *bags, earrings, ties, and shoes*. We used 1800 samples from this dataset. We generated 6 binary classification tasks for each pair of the 4 classes with 200 samples for training, 200 samples for validation, and the rest of the samples for testing performance.

**Neural networks on texts as privileged information:** We used *images* as the *original* domain and *texts* as the *privileged* domain. This setting was also explored in [6]. However, we used a different dataset because textual descriptions of the images used in [6] are sparse and contain duplicates. More precisely, we extracted more advanced text features instead of simple term frequency (TF) features. For the images representation, we extracted SURF descriptors [21] and constructed a codebook of 100 visual words using the $k$-means clustering. For the text representation, we extracted 200 dimensional continuous word-vectors using a neural network skip-gram architecture [22][1]. To convert this word representation into a fixed-length sentence representation, we constructed a codebook of 100 word-vectors using again $k$-means clustering. We note that a more elaborate approach to transform word to sentence or document features has recently been developed [23], and we are planning to explore this in the future. We performed PCA for dimensionality reduction in the original and privileged domains and only kept the top 50 principal components. Finally, we standardised the data so that each feature had zero mean and unit standard deviation.

The experimental results are summarised in Table 1. On average over 6 tasks, SVM with hinge loss outperforms GPC with probit likelihood. However, GPC+ significantly improves over GPC providing the best results on average. This clearly shows that GPC+ is able to employ the neural network textual representation as privileged information. In contrast, SVM+ produced the same result as SVM. We suspect this is due to the fact that that SVM has already shown strong performance on the original image space coupled with the difficulties of finding the best values of the four hyper-parameters of SVM+. Keep in mind that in SVM+ we discretised the hyper-parameter search space over 625 ($5 \times 5 \times 5 \times 5$) possible combination values and used a separate validation set to estimate the resulting prediction performance.

### 4.2 Animals with attributes (AwA) dataset

The dataset was collected by querying image search engines for each of the 50 animals categories which have complimentary high level descriptions of their semantic properties such as shape, colour, or habitat information among others [24]. The semantic attributes per animal class were retrieved from a prior psychological study. We focused on the 10 categories corresponding to the test set of this dataset for which the predicted attributes are provided based on the probabilistic DAP model [24]. The 10 classes are: *chimpanzee, giant panda, leopard, persian cat, pig, hippopotamus, humpback whale, raccoon, rat, seal*, which have 6180 images associated in total. As in Section 4.1 and also in

Table 1: Average error rate in % (the lower the better) on the Attribute Discovery dataset over 100 repetitions. We used **images** as the **original** domain and neural networks word-vector representation on **texts** as the **privileged** domain. The best method for each binary task is highlighted in **boldface**. An average rank equal to one means that the corresponding method has the smallest error on the 6 tasks.

|  | GPC | GPC+ (Ours) | SVM | SVM+ |
|---|---|---|---|---|
| bags v. earrings | 9.79±0.12 | **9.50±0.11** | 9.89±0.14 | 9.89±0.13 |
| bags v. ties | 10.36±0.16 | 10.03±0.15 | **9.44±0.16** | 9.47±0.13 |
| bags v. shoes | 9.66±0.13 | **9.22±0.11** | 9.31±0.12 | 9.29±0.14 |
| earrings v. ties | 10.84±0.14 | **10.56±0.13** | 11.15±0.16 | 11.11±0.16 |
| earrings v. shoes | 7.74±0.11 | **7.33±0.10** | 7.75±0.13 | 7.63±0.13 |
| ties v. shoes | 15.51±0.16 | 15.54±0.16 | **14.90±0.21** | 15.10±0.18 |
| *average error on each task* | 10.65±0.11 | **10.36±0.12** | 10.41±0.11 | 10.42±0.11 |
| *average ranking* | 3.0 | **1.8** | 2.7 | 2.5 |

[6], we generated 45 binary classification tasks for each pair of the 10 classes with 200 samples for training, 200 samples for validation, and the rest of samples for testing the predictive performance.

**Neural networks on images as privileged information:** Deep learning methods have gained an increased attention within the machine learning and computer vision community over the recent years. This is due to their capability in extracting informative features and delivering strong predictive performance in many classification tasks. As such, we are interested to explore the use of deep learning based features as privileged information so that their predictive power can be used even if we do not have access to them at prediction time. We used the standard *SURF* features [21] with 2000 visual words as the *original* domain and the recently proposed *DeCAF* features [25] extracted from the activation of a deep convolutional network trained in a fully supervised fashion as the *privileged* domain. The DeCAF features have 4096 dimensions. All features are provided with the AwA dataset[2]. We again performed PCA for dimensionality reduction in the original and privileged domains and only kept the top 50 principal components, as well as standardised the data.

**Attributes as privileged information:** Following the experimental setting of [6], we also used *images* as the *original* domain and *attributes* as the *privileged* domain. Images were represented by 2000 visual words based on SURF descriptors and attributes were in the form of 85 dimensional predicted attributes based on probabilistic binary classifiers [24]. As previously, we also performed PCA and kept the top 50 principal components in the original domain and standardised the data.

The results of these experiments are shown in Figure 2 in terms of pairwise comparisons over 45 binary tasks between GPC+ and the main baselines, GPC and SVM+. The complete results with the error of each method GPC, GPC+, SVM, and SVM+ on each problem are relegated to the supplementary material. In contrast to the results on the attribute discovery dataset, on the AwA dataset it is clear that GPC outperforms SVM in almost all of the 45 binary classification tasks (see the supplementary material). The average error of GPC over 4500 (45 tasks and 100 repeats per task) experiments is much lower than SVM. On the AwA dataset, SVM+ can take advantage of privileged information – be it deep belief DeCAF features or semantic attributes – and shows significant performance improvement over SVM. However, GPC+ still shows the best overall results and further improves the already strong performance of GPC. As illustrated in Figure 1 **(right)**, the privileged information modulates the slope of the probit likelihood function differently for easy and difficult examples: easy examples gain slope and hence importance whereas difficult ones lose importance in the classification. In this dataset we analysed our experimental results using the multiple dataset statistical comparison method described in [26][3]. The results of the statistical tests are summarised in Figure 3. When DeCAF attributes are used as privileged information, there is statistical evidence supporting that GPC+ *performs best* among the four methods, while when the semantic attributes are used as privileged information, GPC+ still performs best but there is not enough evidence to reject that GPC+ performs comparable to GPC.

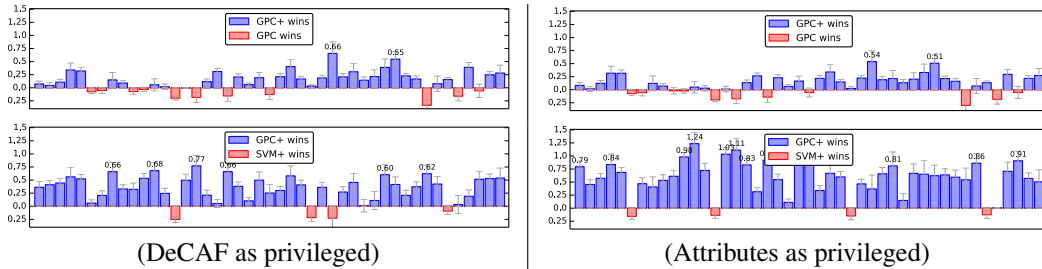

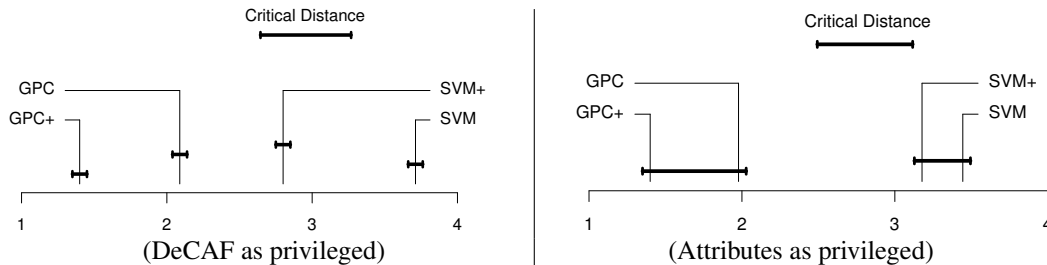

<div align="center">(DeCAF as privileged)　　(Attributes as privileged)</div>

Figure 2: Pairwise comparison of the proposed GPC+ method and main baselines is shown via the relative difference of the error rate (top: GPC+ versus GPC, bottom: GPC+ versus SVM+). The length of the 45 bars corresponds to relative difference of the error rate over 45 cases. Average error rates of each method on the AwA dataset across each of the 45 tasks are found in the supplementary material. (Best viewed in color).

<div align="center">(DeCAF as privileged)　　(Attributes as privileged)</div>

Figure 3: Average rank (the lower the better) of the four methods and critical distance for statistically significant differences (see [26]) on the AwA dataset. An average rank equal to one means that particular method has the smallest error on the 45 tasks. Whenever the average ranks differ by more than the critical distance, there is statistical evidence ($p$-value $< 10\%$) supporting a difference in the average ranks and hence in the performance. We also link two methods with a solid line if they are not statistically different from each other ($p$-value $> 10\%$). When the DeCAF features are used as privileged information, there is statistical evidence supporting that GPC+ performs best among the four methods considered. When the attributes are used, GPC+ still performs best, but there is not enough evidence to reject that GPC+ performs comparable to GPC.

## 5 Conclusions and future work

We presented the first treatment of the *learning with privileged information* paradigm under the Gaussian process classification (GPC) framework, and called it GPC+. In GPC+ privileged information is used in the latent noise layer, resulting in a data-dependent modulation of the slope of the likelihood. The training time of GPC+ is about twice times the training time of a standard Gaussian process classifier. The reason is that GPC+ must train two latent functions, $f$ and $g$, instead of only one. Nevertheless, our results show that GPC+ is an effective way to use privileged information, which manifest itself in significantly better prediction accuracy. Furthermore, to our knowledge, this is the first time that a heteroscedastic noise term is used to improve GPC. We have also shown that recent advances in continuous word-vector neural networks representations [23] and deep convolutional networks for image representations [25] can be used as *privileged* information. For future work, we plan to extend the GPC+ framework to the multi-class case and to speed up computation by devising a quadrature-free expectation propagation method, similar to the ones in [27, 28].

**Acknowledgement:** D. Hernández-Lobato is supported by Dirección General de Investigación MCyT and by *Consejería de Educación CAM* (projects TIN2010-21575-C02-02, TIN2013-42351-P and S2013/ICE-2845). V. Sharmanska is funded by the European Research Council under the ERC grant agreement no 308036.

## Footnotes

[1]`https://code.google.com/p/word2vec/`

[2] http://attributes.kyb.tuebingen.mpg.de

[3] Note that we are not able to use this method on the results of the attribute discovery dataset in Table 1 because the number of methods compared (*i.e.*, 4) is almost equal to the number of tasks or datasets (*i.e.*, 6).

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
