[Supplementary Material · supp_gprivileged.pdf]

# Supplementary Material – Mind the Nuisance: Gaussian Process Classification using Privileged Noise

Table 1: Error rate performance on the AwA dataset over 100 repeated experiments. **SURF** image features as the **original** domain and **DeCAF** deep neural network image features as the **privileged** domain. The best methods for each binary task is highlighted in boldface.

|  | GPC | GPC+ (Ours) | SVM | SVM+ |
|---|---|---|---|---|
| Chimp. v. Panda | 15.93±0.15 | **15.86±0.15** | 16.90±0.15 | 16.22±0.14 |
| Chimp. v. Leopard | 14.74±0.13 | **14.69±0.12** | 15.28±0.15 | 15.10±0.12 |
| Chimp. v. Cat | 16.63±0.12 | **16.52±0.11** | 17.34±0.11 | 16.97±0.11 |
| Chimp. v. Pig | 19.82±0.23 | **19.48±0.26** | 20.52±0.27 | 20.04±0.25 |
| Chimp. v. Hippo. | 18.99±0.11 | **18.67±0.11** | 19.57±0.11 | 19.20±0.11 |
| Chimp. v. Whale | **5.72±0.08** | 5.79±0.08 | 6.22±0.11 | 5.85±0.10 |
| Chimp. v. Raccoon | **19.60±0.13** | 19.65±0.13 | 20.03±0.14 | 19.86±0.13 |
| Chimp. v. Rat | 19.94±0.27 | **19.79±0.26** | 21.53±0.30 | 20.45±0.27 |
| Chimp. v. Seal | 14.10±0.12 | **14.01±0.12** | 14.94±0.12 | 14.34±0.13 |
| Panda v. Leopard | **14.19±0.12** | 14.26±0.13 | 14.85±0.15 | 14.59±0.15 |
| Panda v. Cat | **12.96±0.12** | 12.99±0.12 | 14.26±0.13 | 13.52±0.13 |
| Panda v. Pig | 20.23±0.23 | **20.17±0.22** | 22.54±0.29 | 20.85±0.24 |
| Panda v. Hippo. | 15.47±0.12 | **15.45±0.13** | 16.89±0.17 | 15.70±0.14 |
| Panda v. Whale | 5.00±0.09 | 5.20±0.09 | 5.73±0.11 | **4.95±0.09** |
| Panda v. Raccoon | **17.03±0.13** | 17.04±0.13 | 18.37±0.18 | 17.54±0.14 |
| Panda v. Rat | **17.09±0.24** | 17.27±0.23 | 20.06±0.30 | 18.04±0.25 |
| Panda v. Seal | 14.13±0.13 | **14.01±0.12** | 15.16±0.17 | 14.22±0.13 |
| Leopard v. Cat | 12.08±0.11 | **11.77±0.09** | 11.79±0.08 | 11.82±0.09 |
| Leopard v. Pig | **20.97±0.24** | 21.12±0.24 | 21.88±0.24 | 21.79±0.26 |
| Leopard v. Hippo. | 16.10±0.14 | **15.89±0.13** | 16.36±0.13 | 16.27±0.14 |
| Leopard v. Whale | 4.85±0.07 | **4.79±0.07** | 4.95±0.08 | 4.89±0.07 |
| Leopard v. Raccoon | 26.43±0.17 | **26.24±0.16** | 26.93±0.20 | 26.74±0.19 |
| Leopard v. Rat | **17.62±0.20** | 17.75±0.21 | 18.78±0.25 | 18.00±0.23 |
| Leopard v. Seal | 13.26±0.10 | **13.05±0.10** | 13.33±0.10 | 13.35±0.10 |
| Cat v. Pig | 24.35±0.24 | **23.95±0.22** | 24.52±0.24 | 24.53±0.25 |
| Cat v. Hippo. | 15.02±0.10 | **14.85±0.10** | 15.52±0.12 | 15.26±0.11 |
| Cat v. Whale | 7.66±0.09 | 7.63±0.09 | 7.51±0.09 | **7.41±0.10** |
| Cat v. Raccoon | 15.54±0.11 | **15.35±0.10** | 15.80±0.10 | 15.71±0.13 |
| Cat v. Rat | 34.96±0.30 | 34.30±0.30 | 34.83±0.32 | **34.07±0.32** |
| Cat v. Seal | 18.79±0.14 | **18.58±0.13** | 18.97±0.14 | 18.85±0.13 |
| Pig v. Hippo. | 27.57±0.27 | **27.27±0.27** | 27.75±0.25 | 27.72±0.27 |
| Pig v. Whale | 8.37±0.15 | **8.22±0.16** | 8.51±0.17 | 8.24±0.16 |
| Pig v. Raccoon | 24.45±0.24 | **24.23±0.23** | 24.40±0.25 | 24.34±0.25 |
| Pig v. Rat | 30.00±0.26 | **29.62±0.26** | 30.77±0.30 | 30.22±0.28 |
| Pig v. Seal | 23.91±0.24 | **23.37±0.21** | 24.35±0.23 | 23.79±0.22 |
| Hippo. v. Whale | 14.03±0.12 | **13.80±0.12** | 14.01±0.12 | 14.01±0.10 |
| Hippo. v. Raccoon | 19.31±0.14 | **19.14±0.13** | 19.61±0.15 | 19.52±0.13 |
| Hippo. v. Rat | **21.49±0.27** | 21.82±0.26 | 22.67±0.26 | 22.44±0.27 |
| Hippo. v. Seal | 30.68±0.17 | **30.60±0.18** | 31.52±0.18 | 31.03±0.19 |
| Whale v. Raccoon | 7.92±0.09 | 7.77±0.08 | **7.61±0.09** | 7.68±0.09 |
| Whale v. Rat | **10.98±0.22** | 11.14±0.22 | 11.38±0.24 | 11.17±0.24 |
| Whale v. Seal | 18.57±0.16 | **18.18±0.16** | 18.58±0.18 | 18.37±0.16 |
| Raccoon v. Rat | **25.16±0.27** | 25.22±0.25 | 25.90±0.24 | 25.73±0.25 |
| Raccoon v. Seal | 15.06±0.13 | **14.82±0.13** | 15.43±0.12 | 15.35±0.12 |
| Rat v. Seal | 24.91±0.28 | **24.63±0.27** | 25.24±0.28 | 25.16±0.28 |
| *average ranking* | 2.09 | 1.40 | 3.71 | 2.80 |
| *average error* | 17.60±0.10 | **17.47±0.10** | 18.21±0.11 | 17.80±0.10 |

Table 2: Error rate performance on the AwA dataset over 100 repeated experiments. **SURF** image features as the **original** domain and **attributes** as the **privileged** domain. The best methods for each binary task is highlighted in boldface.

| | GPC | GPC+ (Ours) | SVM | SVM+ |
|---|---|---|---|---|
| Chimp. v. Panda | 15.93±0.15 | **15.85±0.14** | 16.90±0.15 | 16.64±0.15 |
| Chimp. v. Leopard | 14.74±0.13 | **14.72±0.12** | 15.28±0.15 | 15.18±0.12 |
| Chimp. v. Cat | 16.63±0.12 | **16.51±0.11** | 17.34±0.11 | 17.08±0.10 |
| Chimp. v. Pig | 19.82±0.23 | **19.50±0.26** | 20.52±0.27 | 20.34±0.24 |
| Chimp. v. Hippo. | 18.99±0.11 | **18.68±0.11** | 19.57±0.11 | 19.36±0.11 |
| Chimp. v. Whale | 5.72±0.08 | 5.79±0.08 | 6.22±0.11 | **5.63±0.08** |
| Chimp. v. Raccoon | **19.60±0.13** | 19.66±0.13 | 20.03±0.14 | 20.13±0.13 |
| Chimp. v. Rat | 19.94±0.27 | **19.82±0.26** | 21.53±0.30 | 20.22±0.28 |
| Chimp. v. Seal | 14.10±0.12 | **14.03±0.12** | 14.94±0.12 | 14.57±0.15 |
| Panda v. Leopard | **14.19±0.12** | 14.21±0.13 | 14.85±0.15 | 14.82±0.14 |
| Panda v. Cat | **12.96±0.12** | 12.98±0.12 | 14.26±0.13 | 13.96±0.12 |
| Panda v. Pig | 20.23±0.23 | **20.18±0.22** | 22.54±0.29 | 21.41±0.26 |
| Panda v. Hippo. | 15.47±0.12 | **15.44±0.13** | 16.89±0.17 | 16.16±0.17 |
| Panda v. Whale | **5.00±0.09** | 5.20±0.09 | 5.73±0.11 | 5.06±0.08 |
| Panda v. Raccoon | 17.03±0.13 | **17.02±0.13** | 18.37±0.18 | 18.06±0.15 |
| Panda v. Rat | **17.09±0.24** | 17.26±0.23 | 20.06±0.30 | 18.37±0.25 |
| Panda v. Seal | 14.13±0.13 | **13.99±0.13** | 15.16±0.17 | 14.82±0.15 |
| Leopard v. Cat | 12.08±0.11 | 11.81±0.09 | **11.79±0.08** | 12.13±0.11 |
| Leopard v. Pig | **20.97±0.24** | 21.11±0.24 | 21.88±0.24 | 22.03±0.29 |
| Leopard v. Hippo. | 16.10±0.14 | **15.87±0.13** | 16.36±0.13 | 16.41±0.14 |
| Leopard v. Whale | 4.85±0.07 | **4.78±0.07** | 4.95±0.08 | 4.90±0.07 |
| Leopard v. Raccoon | 26.43±0.17 | **26.27±0.17** | 26.93±0.20 | 27.31±0.19 |
| Leopard v. Rat | **17.62±0.20** | 17.67±0.21 | 18.78±0.25 | 18.84±0.24 |
| Leopard v. Seal | 13.26±0.10 | **13.04±0.10** | 13.33±0.10 | 13.38±0.10 |
| Cat v. Pig | 24.35±0.24 | **24.01±0.23** | 24.52±0.24 | 24.68±0.25 |
| Cat v. Hippo. | 15.02±0.10 | **14.87±0.10** | 15.52±0.12 | 15.47±0.11 |
| Cat v. Whale | 7.66±0.09 | 7.63±0.09 | 7.51±0.09 | **7.48±0.09** |
| Cat v. Raccoon | 15.54±0.11 | **15.32±0.10** | 15.80±0.10 | 15.78±0.10 |
| Cat v. Rat | 34.96±0.30 | **34.42±0.29** | 34.83±0.32 | 34.79±0.28 |
| Cat v. Seal | 18.79±0.14 | **18.60±0.13** | 18.97±0.14 | 19.25±0.16 |
| Pig v. Hippo. | 27.57±0.27 | **27.36±0.26** | 27.75±0.25 | 28.17±0.26 |
| Pig v. Whale | 8.37±0.15 | **8.24±0.16** | 8.51±0.17 | 8.38±0.15 |
| Pig v. Raccoon | 24.45±0.24 | **24.24±0.22** | 24.40±0.25 | 24.91±0.22 |
| Pig v. Rat | 30.00±0.26 | **29.67±0.26** | 30.77±0.30 | 30.33±0.29 |
| Pig v. Seal | 23.91±0.24 | **23.41±0.22** | 24.35±0.23 | 24.03±0.23 |
| Hippo. v. Whale | 14.03±0.12 | **13.82±0.12** | 14.01±0.12 | 14.45±0.13 |
| Hippo. v. Raccoon | 19.31±0.14 | **19.15±0.13** | 19.61±0.15 | 19.74±0.16 |
| Hippo. v. Rat | **21.49±0.27** | 21.79±0.26 | 22.67±0.26 | 22.33±0.26 |
| Hippo. v. Seal | 30.68±0.17 | **30.61±0.18** | 31.52±0.18 | 31.47±0.19 |
| Whale v. Raccoon | 7.92±0.09 | 7.79±0.08 | **7.61±0.09** | 7.66±0.08 |
| Whale v. Rat | **10.98±0.22** | 11.16±0.22 | 11.38±0.24 | 11.17±0.22 |
| Whale v. Seal | 18.57±0.16 | **18.28±0.16** | 18.58±0.18 | 18.99±0.18 |
| Raccoon v. Rat | **25.16±0.27** | 25.21±0.25 | 25.90±0.24 | 26.12±0.26 |
| Raccoon v. Seal | 15.06±0.13 | **14.85±0.12** | 15.43±0.12 | 15.42±0.13 |
| Rat v. Seal | 24.91±0.28 | **24.64±0.26** | 25.24±0.28 | 25.14±0.27 |
| *average ranking* | 1.98 | 1.40 | 3.44 | 3.18 |
| *average error* | 17.60±0.10 | **17.48±0.10** | 18.21±0.11 | 18.06±0.11 |