[Reviews · NeurIPS 2014]

Submitted by Assigned_Reviewer_4

Overview:
--

The authors present a method for Gaussian process classification with privileged information (i.e. information that might be available during training, but not at prediction time). The privileges are encoded into the model as data-dependent modulation of the sigmoid slope in the GP likelihood. This GPC+ approach extends existing work that has been done for SVMs (cf. SVM+).

Detailed comments:
--

Pros: This is a rather polished paper; it is easy to follow, provides sufficient background, and includes comparisons to existing methods. This work can be seen as important, because the models with privileged information are commonly encountered in real-world applications.

Cons: The contributions of this paper are incremental, the novelty lies in the idea of modulating the sigmoid likelihood, and extending the privileged noise approach from SVMs to GPC. As suggested in the Conclusion of the paper, devising a quadrature-free method in Sec. 3 would have provided additional originality and contribution.

Additional comments after author rebuttal:
--

I've read through the rebuttal, and I'm very happy with the additional comments provided.
Summary: A well-polished paper with the main contributions being in modulation of the sigmoid likelihood and introducing privileged noise to GP classification.

Submitted by Assigned_Reviewer_13

The paper suggests a new way for incorporating privileged data to the GPC model. The privileged data is used as a way to estimate the noise level of each input location. Both the noise level g and class probability f are learned jointly using EP with a one dimensional quadrature estimation.

In my view this is an elegant way of using GPC with privileged information. If I understand correctly, adding function g() acts as a way to use locations with similar privileged information to better estimate the “noise” in that location.

Quality: The paper is theoretically sound as it extends the GPC framework using an additive GP model. Overall the paper shows a good theoretical background and compares the results of several examples using different datasets. I would also like for the authors to mention something about the runtime complexity of GPC+.

Clarity : Overall, the paper is clearly written. There are a few locations where editing of very long run-on sentences can help. There is also a italic run-away at line 38. The authors should spend less space explaining known ideas such as GPC and EP and explain more about the meaning of g(), e.g., what exactly does it mean hard/simple point, intuition using a simple example. Another interesting question is will the introduction of g actually hurts the model (as seen in a few examples in the results). Another important point that felt a bit swept under the rug was the approximation of Eq. 14. Why does this approximation work, and how well? Even if there is no room in the manuscript this should be discussed in the supplemental.

Originality: The paper is original in the way it uses privilege information as the “noise” in the GPC model.

Significance The paper showed improvements in several problems using very different datasets. My concern is that the improvement in accuracy might not justify the big increase in complexity of using the GPC+ framework. However, the idea behind GPC+ is nice and can allow a new perspective on the problem.
Summary: Interesting GPC approach to the problem of privileged information. Experiments show its usefulness for several real datasets. To improve clarity, more emphasis and discussion is needed on core of the model (the g function).

Submitted by Assigned_Reviewer_32

In this paper, the authors propose a Gaussian Process Classification framework that is able to use privileged information. In particular, the auxiliary variable GPC is used in which the variance of the Gaussian noise is itself modelled via a GP on the privileged information. This has the effect of allowing the privileged information to control the slope of the resulting sigmoid, and hence the confidence with which training instances are considered.

Quality: This is a good quality paper - it addresses a real problem, and proposes and successfully validates a promising solution. I feel that this is a paper that will be of interest to a significant proportion of the NIPS community. My only very minor issue with the paper is that the authors went straight to EP to perform the inference. Using an approximate inference technique clouds the issue a bit - would it not have been possible to use a sampling technique? Or would this have been computationally prohibitive.

Clarity: Very good - the paper is very well written and clear.

Originality: Although similar models have been used for classification (and projection I think?), I cannot recall any similar method being used for classification.

Significance: Possibly slightly incremental, but a paper that will be of interest to many NIPS attendees.
Summary: This paper proposes a Gaussian Process method for classification with privileged information. It uses the auxiliary variable GPC formulation and uses the privileged information to control the variance of the auxiliary Gaussian. Promising performance is shown.
Author Feedback
Author rebuttal: We thank the reviewers for their positive and constructive feedbacks.

We appreciate that the reviewers fully agree that considering extra information available at training time but not at test time is of interest to NIPS community and that our solution based on privileged noise is an elegant way to do that within a Gaussian process framework.

>> Using an approximate inference technique clouds the issue a bit - would it not have been possible to use a sampling technique?

We agree that a sampling technique such as Hamilton Monte Carlo or Elliptical Slice Sampling is a valid alternative for approximate inference. However, it is expected to be computationally more expensive than EP. We used EP with the vision of scaling up the method to larger datasets. The additional reason for choosing EP is based on the good results that this technique has shown in the context of Gaussian processes for classification, both in terms of accuracy and computational cost [16,17].

We want to highlight as well that deriving EP for GPCs with privileged noise is by itself a contribution since so far heteroscedastic GP techniques have been considered for regression only.

>> The approximation of Eq. 14. Why does this approximation work, and how well?

(14) is approximated in our method using numerical quadrature. Since it is a one-dimensional integral, quadrature is nearly un-beatable in terms of accuracy and computational time (see [15], page 8). The derivatives of the log of (14) with respect to the parameters of Q_old can be approximated by introducing the derivative operation inside the integral. As our empirical investigation demonstrates, the approximation of (14) using numerical quadrature is accurate; EP converged without problems in all the datasets investigated in the paper. An inaccurate approximation of (14) would translate most likely into the non-convergence in EP.

>> Will the introduction of g actually hurt the model (as seen in a few examples in the results)

We will revise the manuscript to include more explanations about the g function. Conceptually, for non-informative samples in the privileged space, the value of g for those samples should be equal to a value that would relate to a global noise value, as in standard GPC. Thus, adding g should never hurt, in principle. Proving this theoretically (not only for GPC+ but also for SVM+) is an interesting and challenging research direction. Experimentally, we observe this behaviour in Table 1. GPC+ always improves, most times significantly, in all but one case (ties v. shoes), where the error increases slightly from 15.51 to 15.54. In the experiments that use the DeCAF features as privileged information (see Table 1 in the supplementary material), only in 5 out of 45 cases, the mean error of GPC+ is higher than the mean error of GPC by the amount of 0.15 or higher. Similarly, in the experiments involving attributes as privileged (see Table 2 in supplementary), only in 4 out of 45 cases GPC+ under-performs GPC with the performance gap that is above 0.15. In summary, our empirical results show that the infrequent performance degradations w.r.t. GPC are small and most likely due to the random partitions of the data or to the type-II maximum likelihood optimisation approach used for hyper-parameter optimisation, which is trapped in local optima.

>> The runtime complexity of GPC+

The training time of GPC+ is nearly twice the training time of a standard GPC but in return, the method improves the accuracy in the large majority of cases. The reason for higher training time is that GPC+ has to train two Gaussian processes, one for the f function and one for the g function. The quadrature method introduces some overhead, but this overhead is linear in the number of training instances. Furthermore, in our experiments we have observed that the training time of GPC+ with marginal likelihood optimisation is comparable to the training time of SVM+. The reason is that the grid-search for finding good hyper-parameters in SVM+ is computationally very expensive. We will clarify this in the final version.